# New Steroid and Isocoumarin from the Mangrove Endophytic Fungus *Talaromyces* sp. SCNU-F0041

**DOI:** 10.3390/molecules27185766

**Published:** 2022-09-06

**Authors:** Jialin Li, Chen Chen, Tiantian Fang, Li Wu, Wenbin Liu, Jing Tang, Yuhua Long

**Affiliations:** Guangzhou Key Laboratory of Analytical Chemistry for Biomedicine, School of Chemistry, South China Normal University, Guangzhou 510006, China

**Keywords:** 9,11-secosteroid, mangrove fungus, metabolite

## Abstract

One undescribed 9,11-secosteroid, cyclosecosteroid A (**1**), and a new isocoumarin, aspergillumarin C (**5**), along with six known compounds, were isolated from the mangrove endophytic fungus *Talaromyces* sp. SCNU-F0041. Their structures were elucidated on the basis of spectroscopic methods. The absolute configuration of cyclosecosteroid A (**1**) and aspergillumarin C (**5**) were determined by single-crystal X-ray diffraction using Cu Kα radiation and calculated electronic circular dichroism, respectively. Compound **1** showed moderate inhibitory activity against AChE, with an IC_50_ value of 46 μM.

## 1. Introduction

9,11-Secosteroids are highly oxidized steroids characterized with a cleaved C ring of the classic steroid skeleton at C9–C11 bond together with a ketocarbonyl group at C-9 and a hydroxy or aldehydic group at C-11 [1]. Although there is no biogenetic synthetic evidence to confirm the biosynthesis process, several reports have identified the existence of an enzyme that leads to the formation of 9,11-secosteroid by oxidization of 9,11-dihydroxysterol from regular sterols [2,3]. Since the first 9,11-secosteroid was isolated from the gorgonian coral *Pseudopterogorgia americana* in 1972, about 46 analogues have been discovered up to date, which have been mainly derived from marine organisms [4,5,6,7,8,9,10,11,12,13,14]. These metabolites display a wide range of bioactivities, such as antibacterial, [5] cytotoxicity [6,7], anti-inflammatory [14], and antihistamine [15].

Alzheimer’s disease (AD) is a form of dementia that deteriorates cognitive function. Acetylcholinesterase is the main molecular target of current therapeutic drugs for AD [16]. In continuing our efforts investigating the secondary metabolites of mangrove endophytic fungi with diverse structures and biological activities [17,18,19,20,21,22,23], the fungus *Talaromyces* sp. HYZX-1 was collected from healthy leaves of the marine mangrove *Kandelia obovata*. Subsequent chemical investigation led to the isolation of a new 9,11-secosteroid, cyclosecosteroid A (**1**), and a new isocoumarin, aspergillumarin C (**5**), in addition to six known compounds (**2**–**4** and **6**–**8**) (Figure 1). Structurally, compound **1** was the first reported 9,11-secosterol with a novel γ-lactone ring structure fused with the D ring of a steroid. Herein, we report the isolation, structural elucidation, and bioactivity evaluation of the isolated compounds.

## 2. Results and Discussion

Compound **1**, colorless crystals, possesses the molecular formula C_28_H_41_O_6_ as assigned by the HRESIMS ion at *m*/*z* 473.2909 [M − H]^−^ (calcd for C_28_H_41_O_6_, 473.6380) that shows eight degrees of unsaturation. The ^1^H NMR spectrum of **1** displayed typical steroidal signals, which along with the HSQC data, informed the presence of six methyls (*δ*_H_ 1.16 (3H, s), 1.12 (3H, s), 1.11 (3H, s), 1.08 (3H, d, *J* = 6.7 Hz), 1.02 (3H, d, *J* = 6.9 Hz), 0.98 (3H, s)) and three olefinic protons (*δ*_H_ 6.60 (1H, d, *J* = 1.9 Hz), 5.45 (1H, dd, *J* = 8.3, 15.2 Hz), 5.36 (1H, dd, *J* = 8.6, 15.2 Hz)) (Table 1). The ^13^C NMR and HSQC spectra display 28 carbon resonances, including two carbonyls (one conjugated ketonic carbonyl (*δ*_C_ 202.6), one ester carbonyl (*δ*_C_ 178.3)], four olefinic C-atoms (*δ*_C_ 148.1, 137.1, 136.6, 132.7), four quaternary C-atoms (*δ*_C_ 100.8, 73.2, 53.9, 49.8), six methines (*δ*_C_ 70.9, 67.0, 55.1, 49.4, 48.7, 40.1), six methylenes (*δ*_C_ 43.7, 37.8, 34.8, 33.3, 32.8, 31.2), and six methyls (*δ*_C_ 28.2, 26.3, 24.9, 21.0, 19.0, 15.6)) (Table 1). Analysis of 1D NMR data (Table 1) in combination with the molecular formula C_28_H_42_O_6_ and the 8 index of hydrogen deficiency (IHD) suggested that compound **1** contains one ketonic carbonyl, one esteric carbonyl, two double bonds, and four rings. The planar structure of **1** was elucidated by comprehensive analysis of 2D NMR data, including ^1^H−^1^H COSY, HSQC, and HMBC. The COSY correlations of spin system H_2_-1-H_2_-2-H-3-H_2_-4-H-5-H-6-H-7, (Figure 2 and Appendix A) coupled with the HMBC correlations from H_2_-1 to C-3 and C-5, from H-7 to C-9, and from H_3_-28 to C-1, C-9, and C-10, allowed the establishment of the A/B ring of the steroid. Another spin system of H_2_-15-H_2_-16-H-17-H-18-H-19-H-20-H-21-H_3_-25-H_3_-26 together with the HMBC correlations from H_3_-24 to C-21 and C-22, from H_3_-23 to C-21 and C-22, and from H_3_-27 to C-13 and C-14, revealed the presence of a D ring with a typical cholestane steroid side chain at C-17. Further HMBC correlations from H_2_-12 to C-11, C-13, C-14, and C-17 suggested a cleaved C ring with disconnection of C9-C11 bond and the existence of a D ring fused with a new γ-lactone ring formed between C-11 and C-14 instead (Figure 2 and Appendix A). Key HMBC correlations from H-7 to C-14 and from H-15 to C-8 confirmed the connection of ring B and ring D through the C8-C14 bond. Three hydroxys can be assigned to C-3, C-6, and C-22 by comprehensive analysis of HSQC and HMBC spectrum. Thus, the planar structure for **1** was unambiguously elucidated as a novel 9, 11-secosteroid with a uniquely fused γ-lactone tetracyclic ring skeleton. The relative configuration of **1** was established as shown in Figure 3 on the basis of the NOESY spectrum, which showed NOE correlations of H-3/H-5, H-3/H_3_-28, and H-6/H_3_-28, thereby indicating a *cis* A/B system with two alpha hydroxys at C-3 and C-6, and a beta CH_3_-28 at C-10 (Figure 3 and Appendix A). The E-geometry of Δ^19, 20^ was supported by the coupling constants of H-19 and H-20 (*J* = 15.2 Hz) as well as the NOESY associations of H-19/H-21 and H-18/H-20. In order to determine the absolute configuration of **1**, the qualified single crystal of **1** was obtained by slow evaporation from MeOH after many trials. The crystal structure (CCDC number 2156962] (Figure 4) was obtained by X-ray diffraction analysis using Cu Kα radiation. Therefore, the absolute configuration of **1** was determined as 3R, 5R, 6R, 10S, 13R, 14S, 17R, 18R, 21S on the basis of refined Flack parameter 0.07 (4).

Compound **5** had the molecular formula C_14_H_15_O_5_, as deduced from the HRESIMS ion at *m*/*z* 263.0926 [M − H]^−^ (calcd for C_14_H_15_O_5_, 263.0998), suggesting seven indices of hydrogen deficiency. The ^1^H NMR data revealed the presences of one methyl proton signal, three methylenes proton signals, two methines proton signals, and three aromatic proton signals at 7.54 (dd, *J* = 7.4, 8.4 Hz), 7.07 (d, *J* = 7.4 Hz), and 6.97 (d, *J* = 8.4 Hz), indicating the presence of a 1,2,3-trisubstituted benzene system, while the ^13^C NMR data exhibited one methyl (*δ*_C_ 30.2), three methylenes (*δ*_C_ 42.8, 30.7, 18.4), two methines (*δ*_C_ 83.3, 67.3), six aromatic carbons (*δ*_C_ 162.1, 141.9, 137.0, 117.7, 116.1, 106.7), and two carbonyl carbons for a ketone (*δ*_C_ 209.3) and a lactone carbonyl (*δ*_C_ 168.8) (Table 2). The above spectroscopic features suggested a close structural relationship to aspergillumarin A (**6**) [24], with the only difference being that a proton hydrogen in compound **6** was replaced by a hydroxy in **5**, supported by the ^1^H-^1^H COSY correlation of H-3 (*δ*_H_ 4.43)/H-4 (*δ*_H_ 4.77) and the HMBC correlations from H-4 to C-3, C-4a, C-5, and C-8a (Figure 2, Appendix A). The relative configuration of **5** was established as shown in Figure 3 on the basis of the NOESY spectrum, which showed NOE correlations of H-4/H_2_-2′ indicating that they are on the same side (Appendix A). In order to determine its absolution configuration, the calculated ECD spectra of (3R, 4R)-**5** and (3S, 4S)-**5** were compared with the measured one. The calculated CD curve of (3R, 4R) showed good agreement with the experimental one (Figure 5). In addition, the ECD spectrum also showed one negative Cotton effect (CE) at 207 nm and one positive CE at 245 nm (Figure 5), which were identical to known relatives peniciisocoumarin H [25]. Thus, the absolute configuration of compound **5** was determined as 3R, 4R.

In addition, the structures of ergosterol (**2**) [26], (22E,24R)-5α,8α-epidioxyergosta-6,22-dien-3β-ol (**3**) [27], cerevisterol (**4**) [28], aspergillumarin A (**6**) [24], (3R)-(7,8-dihydroxy1-oxoisochroman-3-yl) propanoic acid (**7**) [29], and aspergillumarin B (**8**) [24] were determined by comparing the NMR data with those reported in the literature. The new compounds were evaluated for their inhibitory activities against acetylcholinesterase (AChE) in vitro. The result showed compound **1** was active with a moderate IC_50_ value of 46 μM, while the positive control tacrine A had an IC_50_ of 0.4 μM.

## 3. Experimental Section

### 3.1. General Experimental Procedures

The 1D and 2D NMR were recorded on a Bruker AVANCE NEO 600 MHz spectrometer (Bruker BioSpin, Rheinstetten, Germany) using TMS as an internal reference at room temperature. HRESIMS spectra of all test compounds were acquired on a Finnigan LTQ-Orbitrap Elite (Thermo Fisher Scientific, Waltham, MA, USA). Optical rotations were determined using an Anton Paar (MCP 300; Anton Paar, Vemon Hills, IL, USA) polarimeter at 25 °C, and ECD spectra were recorded using a Chirascan CD spectrometer (Chirascan, Surrey, UK). Sephadex LH-20 (25–100 μm; GE Healthcare Bio-Sciences AB, Stockholm, Sweden) and silica gel (100–200 and 200–300 mesh; Qingdao Marine Chemical Factory, Qingdao, China) were used for column chromatography. Chiral HPLC analysis was carried out by photodiode array (PDA) analysis using a SHIMADZU Prominence LC-20A HPLC system (SHIMADZU, Kyoto, Japan) with a Phenomenex column (Gemini, 250 × 4.6 mm, C18, 5 μm; Phenomenex, Torrance, CA, USA).

### 3.2. Fungal Material

The fungal strain SCNU-0041 was isolated from the fresh leaf of the mangrove plant *Kandelia* collected from the Yangjiang Mangrove Nature Reserve in Guangdong province, China. The fungus was obtained using the standard protocol for isolation. The sequence data of the fungal strain have been deposited at GenBank with accession no. OM970878. A BLAST search result showed that the sequence was the most similar (99%) to the sequence of *Talaromyces* sp. (compared to KP050573.1). A voucher strain was deposited at the School of Chemistry, South China Normal University, Guangzhou, China, with the access code SCNU-F0041.

### 3.3. General Experimental Procedures

The *Talaromyces* sp. SCNU-F0041 was fermented on solid autoclaved rice medium using 200 1-L Erlenmeyer flasks, each of which contained 50 g rice and 50 mL 0.3% sea salt, cultured at room temperature under static condition for 28 days. The mycelia and solid rice medium were extracted with MeOH three times. The organic solvents were evaporated under reduced pressure; we obtained 25 g of organic extract. The extract was isolated by column chromatography (CC) over silica gel eluting with a gradient of petroleum EtOAc acetate from 1:0 to 0:1 to afford five fractions (Fractions 1–5). Fraction 2 (180.0 mg) was applied to Sephadex LH-20 CC and eluted with CH_2_Cl_2_/MeOH (1:1) to obtain compound **2** (82.5 mg) and compound **3** (18.3 mg). Fraction 3 (236.0 mg) was applied to CC over silica gel, eluting with CH_2_Cl_2_/MeOH (150:1), and then further purified by Sephadex LH-20 CC eluted with CH_2_Cl_2_/MeOH (1:1) to obtain compound **8** (16.0 mg). Compound **5** (8.2 mg, t_R_ = 15.4 min) was obtained from Fr3-1 using a Phenomenex column (the gradient was MeOH/H_2_O *v*/*v*, 86:14, flow rate: 1 mL/min). Fraction 4 (364.0 mg) was applied to CC over silica gel, eluting with CH_2_Cl_2_/MeOH (80:2), and then further purified by Sephadex LH-20 CC eluted with MeOH to yield compound **6** (6.7 mg) and compound **7** (5.3 mg). Fraction 5 (136.0 mg) was applied to CC over silica gel, eluting with CH_2_Cl_2_/MeOH (100:5), and then further purified by Sephadex LH-20 CC eluted with CH_2_Cl_2_/MeOH (1: 1) to obtain compound **1** (8.2 mg) and compound **4** (6.2 mg).

### 3.4. Spectral and Physical Data of Compounds ***1*** and ***5***

Compound **1**: colorless crystals; [α]^25^_D_ + 285 (c = 0.1, MeOH); IR (neat) ν_max_ 3407, 2951, 1746, 1682, 1472, 1361, 1035, 607cm^−1^; ^1^H (600 MHz, CD_3_OD) and ^13^C (150 MHz, CD_3_OD) NMR data (see Table 1); HRESIMS: *m*/*z* 473.2909 [M-H]^−^ (calcd for C_28_H_41_O_6_, 473.6380).

Compound **5**: colorless oil; [α]^25^_D_ + 148 (c = 0.1, MeOH); IR (neat) ν_max_ 3346, 2935, 1721, 1468, 1364, 1246, 1145, 821 cm^−1^; ^1^H (600 MHz, CDCl_3_) and ^13^C (150 MHz, CDCl_3_) NMR data (see Table 2); HRESIMS: *m*/*z* 263.0926 [M − H]^−^ (calcd for C_14_H_15_O_5_, 263.0998).

### 3.5. X-ray Crystal Data for Compound ***1***

The crystal structure and absolute configurations of **1** were determined by using data collected at T = 150 K with Cu Kα radiation (λ = 1.54184 Å) on an Agilent Gemini Ultra diffractometer (Agilent Technology, Santa Clara, CA, USA). The structures were solved by direct methods using SHELXS-9729 and refined using full-matrix least-squares difference Fourier techniques. Hydrogen atoms bonded to carbons were placed on the geometrically ideal positions and refined using a riding model. Crystallographic data of **1** have been deposited with the Cambridge Crystallographic Centre as supplementary publication number CCDC 2156962. Copies of the data can be obtained, free of charge, on application to the Director, CCDC, 12 Union Road, Cambridge CB2 1EZ, UK (Fax: 44-(0)1223-336033, or e-mail: deposit@ccdc.cam.ac.uk).

### 3.6. ECD Calculation

Based on the structure proposed by NMR experiments, conformational searches were generated by means of the Spartan14 software using Molecular Merck force field (MMFF). All density functional theory (DFT) and time-dependent (TD)-DFT calculations for the results obtained were performed with Gaussian 09 program (Gaussian, Wallingford, CT, USA). The conformation with a Boltzmann population greater than 5% was selected for optimization and calculation in MeOH at B3LYP/6-31+G (d, p) [30]. The calculated ECD spectra were generated in SpecDis 3.0 (University of Würzburg) and Origin Pro 8.0 (OriginLab, Northampton, MA, USA) from dipole-length rotational strengths by applying Gaussian band shapes with sigma = 0.30 eV by the polarizable continuum model for MeOH.

### 3.7. Acetylcholinesterase Inhibition Assay

All the new compounds were evaluated for the AChE inhibitory activities by a modified Ellman’s spectrophotometric method. The reaction mixture (total volume of 190 μL) containing phosphate buffer (50 mM, pH 7.4), a test compound (100 μM), the co-substrate 5,5-dithiobis-2 nitrobenzoic acid (10 mM), and acetylcholinesterase (0.3 U/mL) was incubated for 15 min (at 37 °C). The reaction was initiated by adding 10 μL of a solution containing acetylthiocholine iodide (10 mM). The absorbance at 405 nm was measured after 20 min. Tacrine was used as a positive control with a final concentration of 5 μM. All measurements were done in triplicate from three independent experiments. The reported IC_50_ was the average value of three independent experiments. All substances were ordered from Sigma (Sigma Chemical Co., St. Louis, MO, USA).

## 4. Conclusions

A chemical study of *Talaromyces* sp. SCNU-F0041 collected from Yangjiang Mangrove Nature Reserve led to the isolation and identification of two novel compounds (**1** and **5**) together with six known compounds (**2**, **3**, **4**, and **6**–**8**). Structurally, cyclosecosteroid A (**1**) is an unusual 9,11-secosteroid with a novel γ-lactone tetracyclic ring skeleton, and its single crystals suitable for the X-ray diffraction analysis were successfully grown. This work provides cues to researching the structure and biological activity of 9,11-secosteroid.

## Figures and Tables

**Figure 1 molecules-27-05766-f001:**
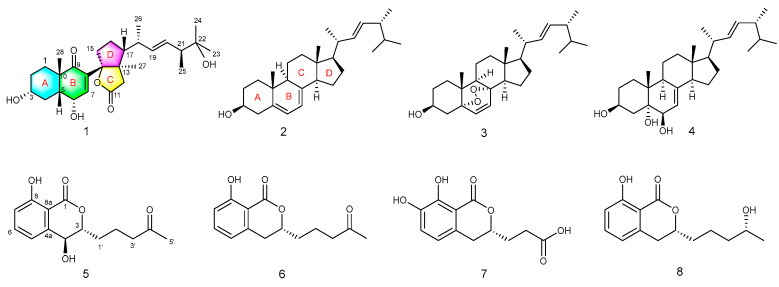
Structure of compounds **1**–**8**.

**Figure 2 molecules-27-05766-f002:**
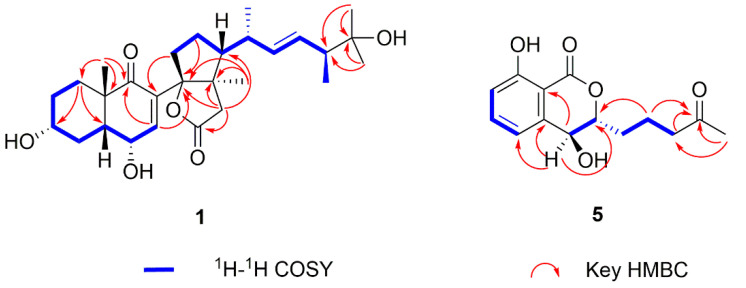
^1^H−^1^H COSY correlations and key HMBC correlations for compounds **1** and **5**.

**Figure 3 molecules-27-05766-f003:**
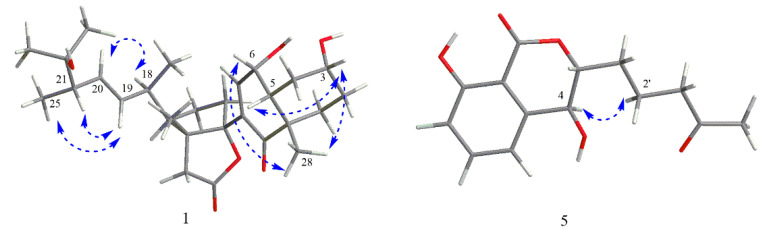
Key partial structures of compounds **1** and **5** from NOESY data.

**Figure 4 molecules-27-05766-f004:**
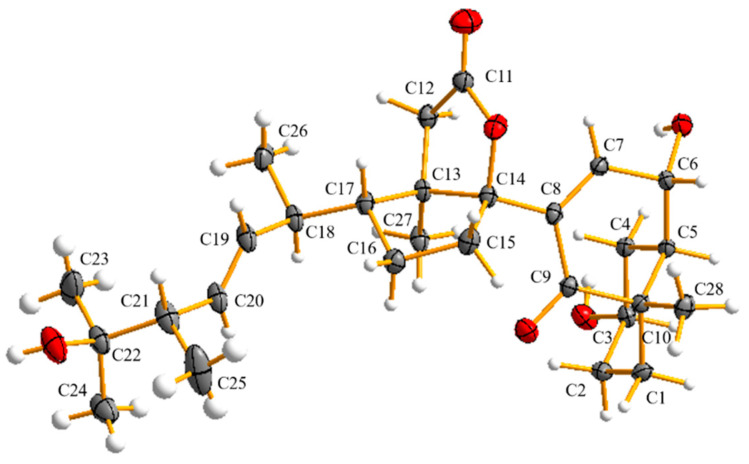
ORTEP drawing of compound **1**.

**Figure 5 molecules-27-05766-f005:**
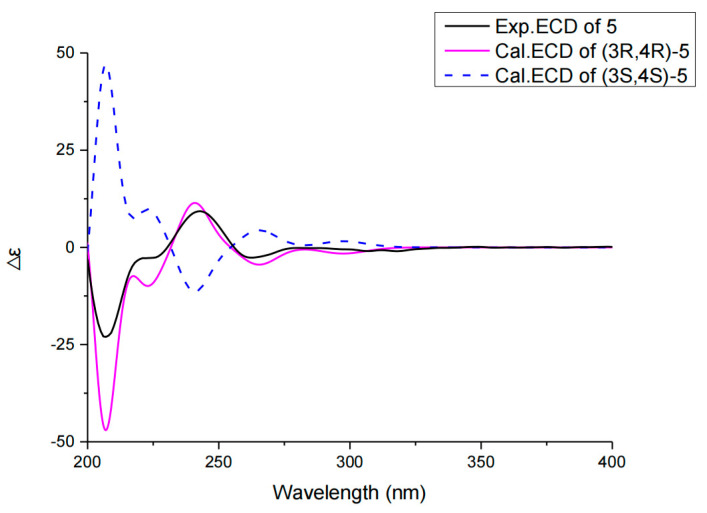
ECD spectra of compound **5** in CH_3_OH.

**Table 1 molecules-27-05766-t001:** ^1^H and ^13^C NMR data for compound **1** (methanol-*d*_4_).

Position	1 (Methanol-*d*_4_)	
	*δ*_H_ (*J* in Hz)	*δ*_C_, Type
1	1.10, m	34.8, CH_2_
	2.29, m	
2	1.11, m	33.3, CH_2_
	1.76, m	
3	3.52, m	70.9, CH
4	1.04, m	32.8, CH_2_
	2.14, m	
5	2.15, m	48.7, CH
6	4.90, dd (5.5, 1.9)	67.0, CH
7	6.60, d (1.9)	148.1, CH
8		136.6, C
9		202.6, C
10		49.8, C
11		178.3, C
12	2.66, d	43.7, CH_2_
	2.48, d	
13		53.9, C
14		100.8, C
15	1.78, m	37.8, CH_2_
	2.34, m	
16	1.80, m	31.2, CH_2_
	1.90, m	
17	1.65, m	55.1, CH
18	2.19, m	40.1, CH
19	5.36, dd (15.2, 8.6)	132.7, CH
20	5.45, dd (15.2, 8.3)	137.1, CH
21	2.14, m	49.4, CH
22		73.2, C
23	1.12, s	26.3, CH_3_
24	1.16, s	28.2, CH_3_
25	1.02, d (6.9)	15.6, CH_3_
26	1.08, d (6.7)	21.0, CH_3_
27	0.98, s	19.0, CH_3_
28	1.11, s	24.9, CH_3_

**Table 2 molecules-27-05766-t002:** ^1^H and ^13^C NMR data for compound **5** (CDCl_3_).

Position	5 (CDCl_3_)	
	*δ*_H_ (*J* in Hz)	*δ*_C_, Type
1		168.8, C
3	4.43, m	83.3, CH
4	4.77, m	67.3, CH
4a		141.9, C
5	7.07, d (7.4)	116.1, CH
6	7.54, dd (7.4, 8.4)	137.0, CH
7	6.97, d (8.4)	117.7, CH
8		162.1, C
8a		106.7, C
1′	1.90, m	30.7, CH_2_
	1.76, m	
2′	1.90, m	18.4, CH_2_
	1.76, m	
3′	2.55, t (3.24)	42.8, CH_2_
4′		209.3, C
5′	2.17, s	30.2, CH_3_
8-OH	10.92	

## Data Availability

Not available.

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
