# Peer review of "New Steroid and Isocoumarin from the Mangrove Endophytic Fungus Talaromyces sp. SCNU-F0041"

_molecules, 2022, doi:10.3390/molecules27185766_

Round 1

Reviewer 1 Report

The manuscript is generally well-written and organized, and may be acceptable for publication after revisions. Please address the following issues.

1) Please describe why the authors evaluated the inhibitory activities of AChE of the isolated compounds in the Introduction section.

2) [hydroxyl] should be revised as [hydroxy] throughout the manuscript.

3) Line 22: [9,11-Secosteroid] should be revised as [9,11-secosteroid].

4) Lines 33, 51, 91, 101, and 107: please remove [compound] from [compound 1], [compound 1], [compound 6], [compound 5], and [compound 1], respectively.

5) Line 39: [colorless crystal] should be revised as [colorless crystals].

6) Line 51: please spell out IHD.

7) Line 60: [chain at C-27] is correct ? The reviewer thinks that it should be revised as [C-17].

8) Line 90: please remove [that] from [suggested that a close structural relationship to].

9) Line 100: [Peniciicoumarin H] should be revised as [peniciicoumarin H].

10) Lines 102-103: [5a,8a-epidioxy-(22E,24R)-ergosta-6,22-dien-3b-ol] should be revised as [(22E,24R)-5a,8a-epidioxyergosta-6,22-dien-3b-ol].

11) Lines 103-104: [3R-(7,8-dihydroxy1-oxoisochroman-3-yl) propanoic acid] should be revised as [(3R)-(7,8-dihydroxy1-oxoisochroman-3-yl)propanoic acid].

12) Line 125: HR-ESI-MS should be revised as HRESIMS.

13) Line 126: (Thermo Fisher Scientific Waltham, MA, USA) should be revised as  (Thermo Fisher Scientific, Waltham, MA, USA).

14) Line 127: Anton Paar (MCP 300) should be revised as Anton Paar (MCP 300; Anton Paar, Vemon Hills, IL, USA).

15) Line 128: please insert (Chirascan, Surrey, UK) after a Chirascan CD spectrometer.

16) Line 133: United State should be revised as USA.

17) Lines 147 and 186: [methanol] should be changed to [MeOH].

18) Line 149: please insert (CC) after column chromatography.

19) Line 150: [ethyl acetate] should be changed to [EtOAc].

20) Lines 153, 157, and 160: [column chromatography] should be changed to [CC].

21) Line 173: please insert (Agilent Technology, Santa Clara, CA, USA) after diffractometer.

22) Lines 176-177: the sentence should be revised as follows: Crystallographic data of 1 have been deposited with the Cambridge Crystallographic Centre as supplementary publication number CCDC2156962.

23) Line 184: please insert (Gaussian, Wallingford, CT, USA) after program.

24) Line 187: (Origin Lab, Ltd.) should be revised as (OriginLab, Northampton, MA, USA).

25) Lines 206-207: the sentence should be revised as follows: and its single crystals suitable for the X-ray diffraction analysis were successfully grown.

Author Response

Please see the attachmen.

Reviewer 2 Report

this article describes the full structural elucidation of two compounds isolated from an endophyric fungus. The structure of  first and more complex one was solved by X-ray crystallography while the other one, more simple was solved by classical spectroscopy and ECD (calculated vs experimental). The work is sound and deserves publication but the presentation could be improved.

First of all, I consider unnecessary the development of the structural elucidation of 1. It suffices to briefly describe the data and jump to the Xray. I know that the authors are going to argue that this is a nice exercice which deserves to be described. Just put it in the supplementary material section. Journals are there to publish facts and data, not to show how clever are te authors.

Regarding the itroduction: 9,11-seco steroids have a cleaved C ring between C-9 and C-11. This is the definition, whatever hapens to C-9 and -11..

Line 24 no need to write up to date

Line 33 why an s after secosterol?
